# Effects of discontinuation of levothyroxine treatment in older adults: protocol for a self-controlled trial

A Janneke Ravensberg [ID],[1] Rosalinde K E Poortvliet [ID],[1,2] Robert S Du Puy [ID],[3] Olaf M Dekkers [ID],[4] Simon P Mooijaart [ID],[5] Jacobijn Gussekloo [ID] [3,5]

¹Department of Public Health and Primary Care, Leiden University Medical Center, Leiden, Netherlands
²University Network for the Care Sector Zuid-Holland, Leiden University Medical Center, Leiden, Netherlands
³Department of Public Health and Primary Care, Leiden University Medical Center, Leiden, Zuid-Holland, Netherlands
⁴Endocrinology and Metabolic Disorders, Leiden University Medical Center, Leiden, Zuid-Holland, Netherlands
⁵Department of Gerontology and Geriatrics, Leiden University Medical Center, Leiden, Zuid-Holland, Netherlands

**Correspondence to**
A Janneke Ravensberg;
a.j.j.ravensberg@lumc.nl

## ABSTRACT

**Background** Many older persons use the thyroid hormone levothyroxine which is often continued for life. Scientifically, there is much uncertainty whether simple continuation is the optimal approach. First, the physical need for levothyroxine can decrease with age thereby posing a higher risk of overtreatment and adverse effects. Second, large trials in subclinical hypothyroidism have shown no benefit for the use of levothyroxine. Interestingly, guidelines do not address re-evaluation of the indication. This self-controlled trial aims to determine the effects of discontinuation of levothyroxine treatment in older adults.

**Methods and analysis** Participants are community-dwelling subjects aged ≥60 years using levothyroxine continuously at a stable dosage of ≤150 µg and a level of thyroid-stimulating hormone (TSH) <10 mU/L. After a control period of 12 weeks, levothyroxine treatment is discontinued gradually using a stepwise approach with regular monitoring of thyroid function guided by their GP. The primary outcome is the proportion of participants withdrawn from levothyroxine while maintaining a free T4 level within the reference range and a TSH level <10 mU/L, 52 weeks after the start of discontinuation. Secondary outcomes are compared with the control period (self-controlled) and include among others, the effects on thyroid-specific and general health-related quality of life. Furthermore, patients' attitudes towards deprescribing and regret regarding discontinuing levothyroxine treatment will be recorded. A total of 513 participants will be recruited to estimate the expected proportion of 50% with a 95% CI ranging from 45% to 55%.

**Ethics and dissemination** Approval was obtained from the institutional Medical Ethics Committee. The Older People Advisory Board Health and Well-being has reviewed the research proposal and their comments were used for improvement. In line with the funding policies of the grant organisation funding this study, the study results will be proactively disseminated to the general public and key public health stakeholders.

**Trial registration number** NL7978; NCT05821881.

## STRENGTHS AND LIMITATIONS OF THIS STUDY

⇒ This is a self-controlled trial to assess the effect of discontinuation of levothyroxine treatment specifically in older adults (aged ≥60 years), using both thyroid function and thyroid-related quality of life as outcome parameters.

⇒ Stepwise dose reduction with regular measurements of thyroid function reduces the risk of prolonged periods of undertreatment potentially resulting in symptoms of hypothyroidism.

⇒ The primary endpoint, successful withdrawal of levothyroxine, is a biochemically defined endpoint, which is not affected by the unblinded design of the study.

⇒ The unblinded design of this study may introduce performance bias and detection bias in the outcomes of the secondary endpoints.

## INTRODUCTION

Levothyroxine is one of the most widely prescribed drugs and is currently ranked in the top 3 of total prescriptions in the USA and UK.[1][2] It is most frequently prescribed and initiated in older adults (aged 50–70 years).[3]

Notably, levothyroxine prescriptions have increased substantially, from 2.3% to 3.5% over the past decades despite a low prevalence (0.1%–1.9%) and a stable incidence of overt hypothyroidism.[2][4] A contributing factor to the rise in levothyroxine prescriptions is likely an increase in the treatment of subclinical hypothyroidism.[5] Subclinical hypothyroidism is a common condition defined by an elevated level of thyroid-stimulating hormone (TSH) and a free T4 (fT4) level within the normal reference range. The prevalence of subclinical hypothyroidism increases with age (5%–20% in men and women older than 60 years).[6] A recent US study showed that most individuals who started on levothyroxine were diagnosed with subclinical hypothyroidism (65.7%).[7] Furthermore, studies indicate that TSH levels at the initiation of treatment are falling,[8][9] which is worrisome because it may increase the chance of ineffective treatment.[10] This holds especially for older adults with subclinical hypothyroidism with TSH levels <10 mU/L since recent randomised clinical trials show that levothyroxine treatment does not improve thyroid-related symptoms,

quality of life, depressive symptoms or cognitive function in older people.[11–13]

Treatment with levothyroxine for both overt and subclinical hypothyroidism, is often continued for life, as current guidelines do not advise re-evaluation of the indication and effect.[8 14] Initial indications for treatment are not always well documented and can be inaccurate or even inappropriate.[15] Patients may have started levothyroxine in the past for a diagnosis made during transient hypothyroidism.[15–17] Furthermore, thyroid (dys-)function varies over time,[18] and thyroid metabolism changes with age.[19] Therefore, it is likely that a considerable proportion of older adults are treated with levothyroxine without a clear indication according to current guidelines. This could be harmful, considering that oversuppression (TSH <0.45 mU/L) has been reported in 41% of older levothyroxine users[20] and increases the risk of bone fractures and atrial fibrillation.[21 22] The risk of oversuppression increases with treatment duration[8] and although guidelines warrant careful medication monitoring, levothyroxine dosages may remain unchanged for a long time.[23]

Taking into account the high levothyroxine use in older age, the controversy regarding treatment of subclinical hypothyroidism, the lack of re-evaluation of indications, and the health risks associated with oversuppression, it can be hypothesised that discontinuation of levothyroxine treatment can be achieved without negative consequences for a considerable proportion of older adults. A meta-analysis based on observational studies including participants aged 2–81 years suggested that approximately 37% remain euthyroid after thyroid hormone discontinuation.[24] However, the evidence was of low quality, the adult participants were mostly middle-aged, and no study followed a systematic approach for deprescribing levothyroxine.

Therefore, the primary aim of this self-controlled trial is to investigate what proportion of community-dwelling levothyroxine users aged 60 years and older can withdraw from levothyroxine treatment successfully. Secondary aims are as follows: (1) to study what proportion of participants can achieve a substantial dose reduction of levothyroxine at 52 weeks (a) defined as ≥50% and (b) defined by the participants themselves; (2) to study the effect of discontinuation of levothyroxine treatment at 52 weeks on (a) thyroid-related quality of life and (b) general health; (3) to determine how participants reflect on their decision to discontinue levothyroxine treatment.

## METHODS
### Study design
This is a self-controlled trial investigating the stepwise discontinuation of levothyroxine treatment in adults aged 60 years and older with a follow-up of 1 year. Outcomes will be measured 12 weeks before the start of discontinuation, at the start of discontinuation, at 6 weeks, at the end of discontinuation and after 52 weeks. The outcomes will be compared within participants. In this way, the research

questions can be studied validly in a self-controlled design, without the need for an extra control group. The formal role of a control group in a randomised trial is to estimate what would have happened in the experimental arm in the absence of the intervention.[25] This effect is immediately clear in our study: levothyroxine treatment is simply continued in almost all patients. Therefore, a control group in our study will only reveal information that is already known.[26] The primary outcome of the study is based on a biochemically defined thyroid function status, an fT4 level within the reference range and a TSH level <10 mU/L. Since this biochemically defined outcome is independent of participants' views or experiences, is not influenced by personal interpretation and is therefore not affected by an unblinded design, we considered a control group as not required. A TSH level of <10 mU/L was based on the results from the recently published randomised clinical trials on levothyroxine treatment in older adults with subclinical hypothyroidism.[11–13]

### Study setting
Community-dwelling participants are recruited from general practitioner (GP) practices in the Netherlands. Participating GPs identify eligible patients for whom they are the primary caregiver concerning levothyroxine treatment, from electronic medical records using the inclusion and exclusion criteria mentioned below. GPs may exclude patients when they feel study participation could harm their patient. Reasons for exclusion are collected. Selected patients receive an invitation to participate in the study and an information leaflet from their GP (see online supplemental appendix 1). Written informed consent is obtained from all patients who are willing to participate in the study (see online supplemental appendix 1). Participants are enrolled by the study centre.

### Study population
A participant must meet all of the following inclusion criteria:
1. Age ≥60 years
2. Using any levothyroxine monotherapy medicament (ATC: H03AA01) at a stable dosage for ≥1 year.
   Exclusion criteria are as follows:
1. Last measurement of TSH ≥10 mU/L during levothyroxine treatment.
2. The dose of treatment is >150 mcg levothyroxine per day (for safety reasons).
3. The current reason for levothyroxine treatment: patients with a history of thyroidectomy; radioactive iodine treatment or neck irradiation; congenital hypothyroidism; secondary hypothyroidism or concurrent amiodarone or lithium use.
4. Concurrent treatment with liothyronine (the synthetic form of thyroid hormone T3), thiamazole, carbimazole or propylthiouracil (inhibitors of T3 and T4 synthesis).
5. Diagnosis of heart failure NYHA (New York Heart Association) grade IV.

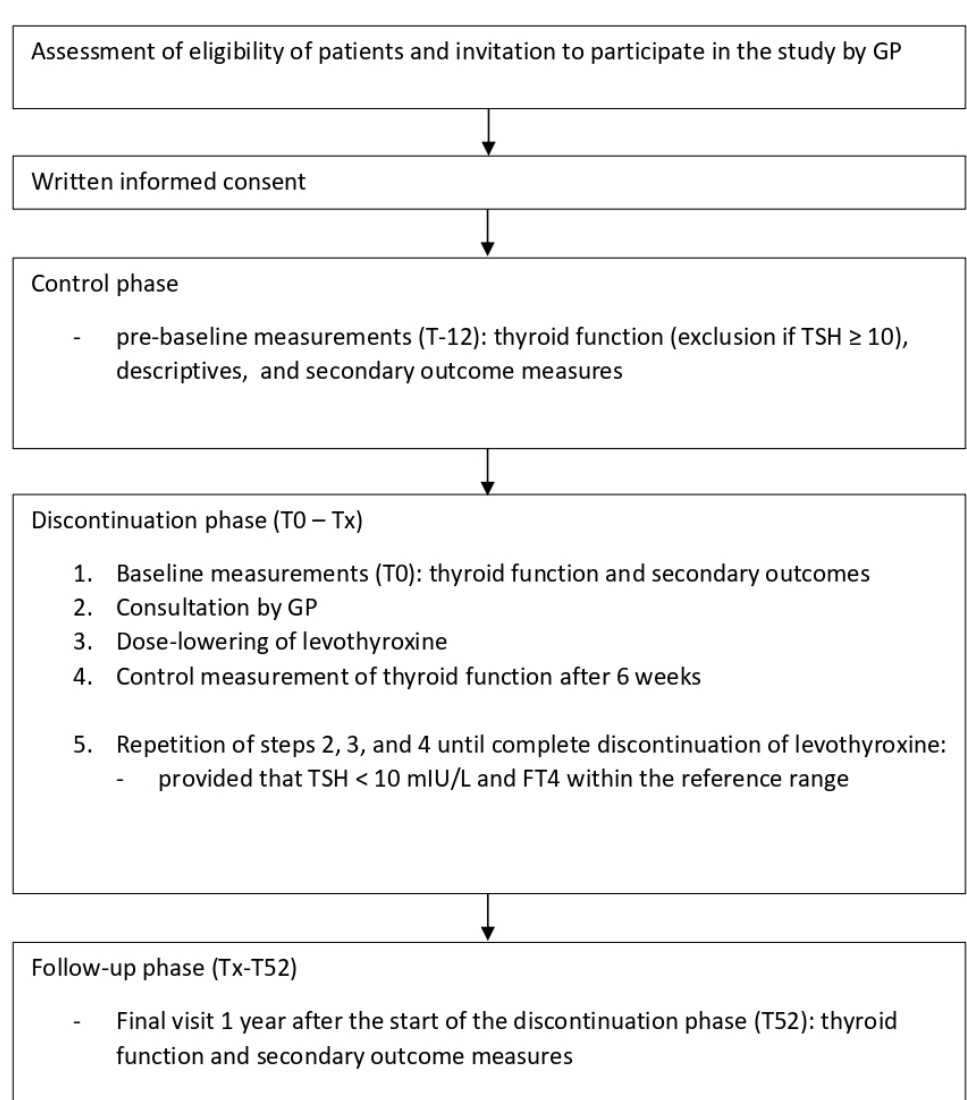

**Figure 1** Flow of participants.

6. Participation in ongoing trials of therapeutic interventions.
7. Life-expectancy<6 months.
8. Diagnosis of dementia.
9. Incapacitated adults.
10. Persons planning to move out of the region in which the study is being conducted in the next months.

### Study phases and interventions

This study consists of three phases: a control phase of 12 weeks (T-12 to T0), the discontinuation phase (minimum of 6 weeks, maximum of 36 weeks; T0 to Tx) and a follow-up phase (from the end of discontinuation till 52 weeks after the start of discontinuation; Tx to T52) (see figure 1).

### Control phase

Participants enter the study during a control period with prebaseline measurements of thyroid function and secondary outcome parameters to determine natural variation after which treatment with levothyroxine is continued per usual care for at least 12 weeks (see table 1). Participants with a prebaseline TSH level ≥10mU/L do not proceed to the discontinuation phase (exclusion). When a participant presents with a suppressed TSH (indicative of oversuppletion with levothyroxine), the discontinuation phase can be started immediately. Participants are requested to complete their questionnaires prior to thyroid function assessment, so the results do not influence their answers. Furthermore, they are instructed to schedule all laboratory visits at the same time preferably in the morning before ingesting levothyroxine and consumption of food and/or drinks.

### Discontinuation phase

The discontinuation phase starts with baseline measurements of thyroid function and secondary outcome parameters (see table 1), followed by a consultation of the participant with their GP during which discontinuation of levothyroxine is initiated, except when TSH level ≥10mU/L. Levothyroxine is discontinued stepwise, and

**Table 1** Schedule of measurements

| | Study phase | | | | | Follow-up phase |
| | Control phase | Discontinuation phase | | | | |
| | | Start of step 1 | Control of step 1 | Control of steps 2–5 | Control of last step‡ | |
|---|---|---|---|---|---|---|
| Time (weeks) | −12 | 0 | 6 | 12, 18, 24, 30 | 6/12/18/24/30/36 | 52 |
| Primary outcome | | | | | | |
| TSH, fT4 | X | X | X | X† | X | X |
| Secondary outcomes | | | | | | |
| ThyPRO-39 | X | X | X | | X | X |
| TSQM | X | X | | | | |
| EQ5D VAS | X | X | X | | X | X |
| DRS | | | | | | X |
| GAS* | | | | | | X |
| Descriptives | | | | | | |
| Demographics and clinical characteristics | X | | | | | |
| ISCOPE | X | | | | | X |
| rPATD | X | X | | | | |
| GAS* | X | | | | | |

*Substantial dose reduction of levothyroxine.
†If applicable.
‡End of the discontinuation phase.
DRS, Decision Regret Scale; EQ5D, EuroQol 5 Dimensions; fT4, free T4; GAS, Goal Attainment Scale; ISCOPE, Integrated Systematic Care for Older Persons; rPATD, revised Patients' Attitudes Towards Deprescribing; ThyPRO, Thyroid-Related Quality of Life Patient-Reported Outcome; TSH, thyroid stimulating hormone; TSQM, Treatment Satisfaction Questionnaire for Medication; VAS, Visual Analogue Scale.

at intervals of ≥6 weeks to ascertain a steady-state concentration has been reached before the following control visit. This is in line with standard clinical practice. During each '6 week' visit, the GP evaluates thyroid function and symptoms with the participant. On agreement, the daily dose of levothyroxine can then be lowered provided thyroid function is within the predefined limits, that is, a normal fT4 and TSH <10 mU/L. With each step, the dose of levothyroxine is reduced by approximately 25 mcg (the use of whole tablets is preferred) until a daily dose of 25 mcg or less is reached. The GP is free to adjust the size of the dose reduction if necessary. The maximum dose reduction in medication is 50 mcg in the first step. In the final step, the medication is discontinued completely if the laboratory results are still within the predefined limits. The discontinuation phase ends after the evaluation of complete withdrawal of the levothyroxine treatment (ie, 6 weeks after the final discontinuation step). Secondary outcome parameters are collected at the first '6 week' visit and the end of discontinuation (see table 1).

Participants are requested to stop the discontinuation phase when a TSH level ≥10 mU/L or an fT4 level below the normal range is identified during a control visit and is confirmed by an extra thyroid function measurement within 2 weeks. When the extra measurement does not confirm a TSH level ≥10 mU/L or a low fT4 level, the participant is allowed to proceed with the discontinuation phase. Factors other than thyroid function, such as symptoms, may however be a reason to prematurely stop the discontinuation or restart levothyroxine. On stopping the discontinuation phase, it is at the discretion of the physician and the participant to decide whether the dose of levothyroxine at that moment will be maintained or increased. Participants who have stopped the discontinuation phase prematurely will be followed up in accordance with the protocol if possible.

### Follow-up phase
The follow-up phase ends with a final visit 52 weeks after the start of the discontinuation phase and includes the assessment of thyroid function, current dose of levothyroxine, and secondary outcome parameters (see table 1).

### Adherence
Participants are contacted by the research nurse by telephone and/or by mail or email prior to each measurement and in case of missing data, to ensure data quality and to promote participant retention and complete follow-up.

### Descriptive study data and outcome measures
Descriptive data recorded at prebaseline include (see online supplemental appendix 2):
1. Age and sex.
2. Lifestyle: smoking, alcohol intake.

3. Medical history.
4. Complex health problems (Integrated Systematic Care for Older Persons (ISCOPE) questionnaire).[27 28]
5. The participant's attitude towards deprescribing (revised Patients' Attitudes Towards Deprescribing (rPATD) questionnaire).[29]
6. Treatment satisfaction for levothyroxine treatment (Treatment Satisfaction Questionnaire for Medication (TSQM)).[30]
7. The participant's view on what is a substantial dose reduction of levothyroxine (Goal Attainment Score (GAS)).

The primary outcome is the proportion of participants who withdraw their thyroid medication successfully (defined as normal fT4 levels and TSH levels <10mU/L) at 52 weeks after the start of the discontinuation (final follow-up).

Secondary outcome measures determined at baseline and/or final follow-up include (see online supplemental appendix 2):

► Thyroid-related quality of life (ThyPRO-39).[31 32]
► General health-related quality of life (EuroQol-5D, including the EuroQol visual analogue scale.[33] Cognitive function is determined as an additional dimension (EuroQol-5D+C).[34 35]
► Participant's view on whether they achieved a substantial dose reduction of levothyroxine (Goal Attainment Scale (GAS)).
► Participant's view on regret after the decision to withdraw levothyroxine (Decision Regret Scale (DRS)).[36]

To further explore the quality of life during the discontinuation phase, ThyPRO-39 and EuroQol measurements were also determined 6 weeks after the first step of lowering the dose of levothyroxine (T6) and 6 weeks after the last step of dose-lowering (see table 1).

## Sample size calculation

The sample size calculation is based on the primary outcome, the proportion of participants successfully stopping levothyroxine treatment between baseline and final follow-up at 52 weeks. Since population proportions of successful levothyroxine treatment cessation are unknown (the aim of this study), we estimated a proportion of 50% of the participants who successfully discontinue levothyroxine treatment. An estimated proportion of 50% results in the most conservative estimate (resulting in the highest calculated sample size) and is in line with the limited available evidence.[37–39] We have chosen a sample size of 385 participants because this allows us to estimate the expected proportion of 50% with a 95% CI of which the lower and upper limit is expected to deviate by 5 percentage points from the found percentage (ie, 45% to 55% when the found percentage is indeed 50%).

Taking a maximum of 25% loss to follow-up into account over the duration of 15 months (exclusion at T-12 when TSH level ≥10mU/L and/or fT4 below reference range, withdrawal, moving away or death between T-12 and T52) a maximum of 513 participants will be recruited. Based on a data query from the academic network of general

practices in the region of Leiden, it was estimated that the projected sample size will be reached after inclusion of 53 standard general practices.

## Data analysis plan

GPs transfer participants' data after each visit using a validated electronic data capture (EDC) system (Castor EDC). Questionnaires are completed on paper or electronically (Castor EDC). All participant data are coded. Only the principal investigators and the research team have access to the key to these codes. The handling of personal data complies with the European General Data Protection Regulation. Database validation checks are run routinely and are tracked and escalated as appropriate.

To determine the proportion of the population that can be withdrawn from levothyroxine treatment, categorical data involving measurements at baseline and final follow-up will be analysed. Univariable and multivariable logistic regression analyses will be performed to assess factors that are associated with successful discontinuation of levothyroxine treatment. To estimate the effect of discontinuation on secondary outcome parameters measured at baseline and/or final follow-up, descriptive, univariable and multivariable analyses will be performed for the overall group and for the group of participants who successfully discontinued levothyroxine. Measurements pre and post discontinuation will be compared. Variations in thyroid function, thyroid-related quality of life, treatment satisfaction for levothyroxine, general health and participant's attitude towards deprescribing will be determined using prebaseline and baseline measurements. All analyses will be carried out with the population of participants who enrolled at baseline (T0). Participants who experience intercurrent events not related to the study procedure resulting in missing data at final follow-up will be excluded. Primary and secondary analyses will be repeated with all participants who enrolled at baseline who have data for the outcome variable of interest and have had their final follow-up visit. Statistical significance for the primary outcome requires p ≤0.05.

## Safety and monitoring
### Withdrawal

Participants can withdraw their consent for involvement in the study at any time, without an obligation to justify the decision and without consequences for their usual care. The research team can advise withdrawing participants from the intervention if it is deemed to be in the best interest of the participant. Participants who withdraw from the study are not replaced. The Data Safety and Monitoring Board (DSMB) can advise to stop the study.

### (Serious) adverse events

To avoid prolonged periods of thyroid hormone under-replacement potentially resulting in symptoms of hypothyroidism, levothyroxine treatment is discontinued gradually while monitoring thyroid function after each reduction. Within the ThyPRO-39, symptoms of

hypothyroidism are assessed as a possible adverse effect. All adverse events (including cardiovascular events) reported spontaneously by the participant or their GP are recorded and followed up. If complaints are reported during the discontinuation phase, these will be assessed by the research nurse. It is at the discretion of the GP to decide to either continue the withdrawal of levothyroxine or to maintain or increase the dosage. Serious adverse events are sought out actively and reported to the sponsor, the accredited Medical Ethics Committee (MEC), and competent authority using electronic case report forms.

## Monitoring

At regular intervals, the DSMB advises the study project group and sponsor (Leiden University Medical Center (LUMC)) whether it is safe and appropriate to continue the study. Periodic routine reports on the progress of the study are sent to the DSMB for review. Serious adverse events are reported immediately. No interim analyses for efficacy or futility will be performed. The DSMB meets at least five times and is composed of medical experts and a biostatistician without any involvement in the study as investigators or as study participant care physicians. The DSMB members are professor J Wouter Jukema (Chair; Department of Cardiology, LUMC), Dr. Marieke Snel (Department of Internal Medicine—Endocrinology, LUMC) and Dr. Ir. Nan van Geloven (Department of Biomedical Data Sciences, LUMC).

Annual study monitoring visits are conducted by an independent clinical research associate (LUMC) according to a study-specific monitoring plan.

A summary of the study progress is submitted to the MEC yearly, including the date of inclusion of the first participant, the number of participants included, the number of participants that have completed the trial, serious adverse events and amendments.

Roles and responsibilities (committees) are also described in online supplemental appendix 3, as is additional trial information regarding trial registration data and trial sponsor.

## Patient and public involvement statement

An online survey (July 2017) revealed that 92% of Dutch GP practices find that investigating the effects of levothyroxine treatment discontinuation is relevant to primary care. The Older People Advisory Board Health and Well-being has carefully reviewed the research proposal and their comments were used for improvement. In a letter of recommendation, the Older People Advisory Board Health and Well-being has given its full support and endorsement. The results will be discussed with the Older People Advisory Board Health and Well-being as well.

## ETHICS AND DISSEMINATION

This study will be conducted according to the principles of the Declaration of Helsinki (amended most recently in October 2013), in accordance with the Dutch Medical Research Involving Human Subjects Act (WMO), the Guideline for Good Clinical Practice (May 1996) and in full conformity to any applicable state or local regulations. In accordance with Dutch legal requirements (WMO), participants will be insured by an insurance of the LUMC to provide cover for damage to research participants through injury or death caused by the study (see online supplemental appendix 1). The protocol has been approved by the accredited MEC (Medisch Ethische Toetsingscommissie Leiden-Den Haag-Delft). Any protocol modifications will be communicated to all relevant parties (including the accredited MEC, the DSMB, participating GPs, participants and trial register).

To comply with the general social responsibility associated with clinical research, the study results will be proactively disseminated to the general public and key public health stakeholders including Schildklier Organisatie Nederland, the patient advocacy group. The findings will be submitted for peer-reviewed publication in national and international medical journals. Publications will be approved by the principle investigators. Individual deidentified participant data can be shared on request only, when the participant has given permission (ie, informed consent) for data sharing concerning future research regarding thyroid disease and treatment with levothyroxine. The study protocol and the statistical analysis plan will be made available online before publication of the main results from this study.

**Contributors** Concept and design: AJR, RKEP, RSDP, OMD, SPM and JG. Drafting of the manuscript: AJR, RKEP and JG. Critical revision of the manuscript: RSDP, OMD and SPM. All authors read and approved the final version. All authors agree to be accountable for all aspects of the work in ensuring that questions related to the accuracy or integrity of any part of the work are appropriately investigated and resolved.

**Funding** This work was supported by ZonMw, programme 'Primary Care and Elderly Care', grant number 839110026. The funder had no role in the design and conduct of the study; collection, management analysis and interpretation of the data; preparation, review or approval of the manuscript or the decision to submit the manuscript for publication.

**Competing interests** Ravensberg receives a grant from ZonMw (839110026) during the conduct of the study.

**Patient and public involvement** Patients and/or the public were involved in the design, or conduct, or reporting or dissemination plans of this research. Refer to the Methods section for further details.

**Patient consent for publication** Not required.

**Provenance and peer review** Not commissioned; externally peer reviewed.

**ORCID iDs**
A Janneke Ravensberg http://orcid.org/0000-0001-8360-6742
Rosalinde K E Poortvliet http://orcid.org/0000-0001-6521-8770
Robert S Du Puy http://orcid.org/0000-0001-8909-7686
Olaf M Dekkers http://orcid.org/0000-0002-1333-7580
Simon P Mooijaart http://orcid.org/0000-0003-3106-3568
Jacobijn Gussekloo http://orcid.org/0000-0001-7186-8278

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
