## [Reviewer comments · BMJ Open]

ARTICLE DETAILS

TITLE (PROVISIONAL)	The effects of discontinuation of levothyroxine treatment in older adults: Protocol for a self-controlled trial
AUTHORS	Ravensberg, A.; Poortvliet, Rosalinde K. E.; Du Puy, Robert; Dekkers, Olaf; Mooijaart, Simon; Gussekloo, Jacobijn

VERSION 1 – REVIEW

REVIEWER	Yilmaz, Ozlem Istanbul Universitesi
REVIEW RETURNED	04-Jan-2023

GENERAL COMMENTS	The issue of treatment decision for hypothyroidism in the elderly is frequently encountered in clinical practice. The problem of when treatment is beneficial and when to discontinue it is a gray zone. In this context, the study tries to enlighten us on this subject. Thank you to the authors. I would like to make a few suggestions to the authors. The findings and discussion summary that should be included in the abstract and main manuscript section should be clearly stated. Even if the study has not yet been concluded, it would be meaningful to include information about the available data.
---

REVIEWER	Brito, Juan Mayo Clinic
REVIEW RETURNED	10-Jan-2023

GENERAL COMMENTS	Janneke Ravensberg and colleagues report the protocol of a study aimed at determining the effect of levothyroxine discontinuation in older adults. Please see the comments below as suggestions to improve the protocol and study. Line 41-42. The sentence is not clear; what do you mean by “re-evaluating” and the end of the sentence? The introduction makes several mentions of overtreatment; however, it is unclear what levothyroxine overtreatment is. For instance, Line 44 “considering that overtreatment occurs in up to 40% of levothyroxine users”. Are these older patients with mild subclinical hypothyroidism receiving levothyroxine? Or older patients receiving levothyroxine without clear indications (euthyroidism, or without the correct diagnosis of subclinical hypothyroidism -not repeating TSH levels-)? Are there any preliminary data about study feasibility, particularly about recruitment and retention and acceptability of levothyroxine discontinuation?
--

	It is unclear if the researchers will include patients receiving levothyroxine for SCH or patients with SCH during treatment. Two different subgroups of patients. The trial will likely include patients willing to discontinue; this will likely be enriched with patients taking levothyroxine without symptoms affecting the generalizability of findings. How will the authors include patients with symptoms at baseline? How will the research prevent that other providers from measuring thyroid levels outside the study (making non-study-approved levothyroxine adjustments) Weight changes is a major concern of patients receiving levothyroxine, please explain why this was not considered. Placebo and randomization are essential to understand the comparative effect of discontinuation on quality of life and symptoms. How will the assessment of symptoms and quality of life be affected in this self-controlled trial? Please explain the rationale for choosing a 150 mcg dose. It seems high, given that an amount of > 100 mcg is unlikely to be used for SCH. Also, a dose of 150 mcg for a patient's weight < 50 mg is different than that of 150 mcg for a patient weighing > 50 mcg. Consider excluding patients with recent hospitalizations and patients with severe dyslipidemia and severe depression How will adherence to levothyroxine assess at baseline and during the study? Please explain the rationale for using the chosen measures. Why was ThyPRO39 chosen? How will symptoms be assessed during the discontinuation phase? And what are the changes in symptoms that will triggered stopping the trial for an individual patient? Will cardiovascular events be assessed as a safety outcome during the trial? In the analysis, please expand on the subgroup of interest and how the heterogeneity of effect will be assessed. Is the calculated sample size large enough to estimate differences for other outcomes? Several observational discontinuation studies found that thyroid parenchyma ultrasound appearance strongly predicted discontinuation. Is this variable considered in this study? If not, why?
--	---

VERSION 1 – AUTHOR RESPONSE

Reviewer: 1

Dr. Ozlem Yilmaz, Istanbul Universitesi

Comments to the Author:

Dear editor

The issue of treatment decision for hypothyroidism in the elderly is frequently encountered in clinical practice. The problem of when treatment is beneficial and when to discontinue it is a gray zone. In this context, the study tries to enlighten us on this subject. Thank you to the authors. I would like to make a few suggestions to the authors.

- The findings and discussion summary that should be included in the abstract and main manuscript section should be clearly stated. Even if the study has not yet been concluded, it would be meaningful to include information about the available data.

In line with the BMJ Open submission guidelines for a protocol paper (<https://bmjopen.bmj.com/pages/authors#protocol>), we have structured the abstract with the following sections: Introduction; Methods and analysis; Ethics and dissemination. We have included registration details as a final section. The primary and secondary outcomes are mentioned in the Methods and analysis section. In accordance with the submission guidelines and as advised by the editor, no preliminary results are mentioned. Strengths and weakness can be found in a separate section in the manuscript.

Reviewer: 2
Dr. Juan Brito, Mayo Clinic

Comments to the Author:

Janneke Ravensberg and colleagues report the protocol of a study aimed at determining the effect of levothyroxine discontinuation in older adults. Please see the comments below as suggestions to improve the protocol and study.

- Line 41-42. The sentence is not clear; what do you mean by “re-evaluating” and the end of the sentence?

If the treatment indication for levothyroxine use would be evaluated according to current guidelines, in older adults who already are receiving levothyroxine treatment, it is likely that a considerable proportion of these patients would not have a clear indication for levothyroxine use.

Adaptation Introduction, page 4, line 25: ‘when re-evaluating’ is deleted from the sentence.

o ‘Therefore, it is likely that a considerable proportion of older adults are treated with levothyroxine without a clear indication according to current guidelines.’

- The introduction makes several mentions of overtreatment; however, it is unclear what levothyroxine overtreatment is. For instance, Line 44 “considering that overtreatment occurs in up to 40% of levothyroxine users”. Are these older patients with mild subclinical hypothyroidism receiving levothyroxine? Or older patients receiving levothyroxine without clear indications (euthyroidism, or without the correct diagnosis of subclinical hypothyroidism -not repeating TSH levels-?

Thank you for your comment. We agree that overtreatment is not the correct wording and not clearly specified. Therefore, we have specified overtreatment as oversuppression as described by Somwaru et al. (line 25-26) and Taylor et al. (line 28-29), or as ineffective treatment as described by Ross (line 12-14).

Adaptations Introduction, page 4, lines:

- o 12-14: ‘Furthermore, studies indicate that TSH levels at the initiation of treatment are falling (8, 9), which is worrisome because it may increase the chance of ineffective treatment (10).’
- o 25-27: ‘This could be harmful, considering that oversuppression (TSH < 0.45 mU/L) has been reported in 41% of older levothyroxine users (21) and increases the risk of bone fractures and atrial fibrillation (22, 23)’
- o 28-29: ‘The risk of oversuppression increases with treatment duration (8) and although guidelines warrant careful medication monitoring, levothyroxine dosages may remain unchanged for a long time (24).’

- Are there any preliminary data about study feasibility, particularly about recruitment and retention and acceptability of levothyroxine discontinuation?

We conducted an online survey among 122 general practices in The Netherlands which showed that 92% of general practitioners thought that investigating the effects of discontinuation of levothyroxine treatment in older adults was relevant to primary care; 81% believed this to be important; 80% felt the research question was currently unanswered; 76% indicated willingness to participate in the current project.

A data query over 2015 and 2016 (from the academic network of general practices in the region of Leiden) showed 566 levothyroxine users among 10400 patients aged 60 or above. In theory 491 of these levothyroxine users could be eligible for participation. It is estimated that the projected sample size will be reached after inclusion of 53 standard general practices We have no preliminary data about retention and acceptability of levothyroxine discontinuation.

Adaptations Methods -sample size calculation page 11, line 14-16.

o Based on a data query from the academic network of general practices in the region of Leiden, it was estimated that the projected sample size will be reached after inclusion of 53 standard general practices.

- It is unclear if the researchers will include patients receiving levothyroxine for SCH or patients with SCH during treatment. Two different subgroups of patients.

In this study, we do not specifically focus on older adults with subclinical hypothyroidism only since also older adults with other indications for levothyroxine treatment may benefit from discontinuation. This may not have been clear from the introduction. Therefore, we have made some adjustments. Both patients receiving levothyroxine for SCH or patients with SCH during treatment can be included in this study, as long as the inclusion and exclusion criteria are met.

Adaptations Introduction, page 4, line 7-8 (merge of paragraph 1 and 2); page 4 line 18.

o 'Treatment with levothyroxine for both overt and subclinical hypothyroidism, is often continued for life, as current guidelines do not advise re-evaluation of the indication and effect (8, 14).'

- The trial will likely include patients willing to discontinue; this will likely be enriched with patients taking levothyroxine without symptoms affecting the generalizability of findings. How will the authors include patients with symptoms at baseline?

Indeed, it is likely that patients willing to discontinue their treatment with levothyroxine will be included in this trial. Whether levothyroxine users exhibit symptoms or not, is not an inclusion criterium. However, we can describe symptoms at baseline using the ThyPRO39, and stratification can give insight whether symptoms are associated with successful discontinuation of levothyroxine or not.

- How will the research prevent that other providers from measuring thyroid levels outside the study (making non-study-approved levothyroxine adjustments)

The participant's GP is the primary care giver concerning the levothyroxine treatment. Participants are instructed that adjustments in levothyroxine treatment are made by their GP. They are also instructed to inform the study team if other care providers become involved in the levothyroxine treatment.

Adaptation Methods-study design, page 6, line 21-23

o 'Participating GPs identify eligible patients for whom they are the primary care giver concerning levothyroxine treatment, from electronic medical records using the in- and exclusion criteria mentioned below.'

- Weight changes is a major concern of patients receiving levothyroxine, please explain why this was not considered.

Weight is not included as a secondary endpoint. Since no home visits are scheduled, it is not possible to perform standardized measurements of weight changes. However, adverse events will be recorded, this could also be weight changes. The only question related to weight change is question 12a in the ThyPRO39.

- Placebo and randomization are essential to understand the comparative effect of discontinuation on quality of life and symptoms. How will the assessment of symptoms and quality of life be affected in this self-controlled trial?

The formal role of a control group in a randomized trial is to estimate what would have happened in the experimental arm in the absence of the intervention [Hernan, M.A. and J.M. Robins, Causal Inference. 2018]. This effect is immediately clear for our study: levothyroxine treatment would simply be continued in almost all patients. Therefore, in our study a control group would only reveal information that is already known [Glasziou, P., et al., When are randomised trials unnecessary? Picking signal from noise. BMJ, 2007. 334(7589): p. 349-51]. The primary outcome of the study (successful discontinuation) is based on biochemically defined thyroid function status (normal fT4 levels and TSH levels <10 mU/L). This outcome is independent of participants' views or experience, is not influenced by personal interpretation and therefore does not require a control group. However, due to the unblinded design, we cannot exclude the occurrence of performance bias and/or detection bias influencing the secondary outcome parameters as stated in 'strengths and limitations' (page 3). The participants are of course aware of every dose-lowering step during the study. Yet, we will ensure that they are blinded to their thyroid function status when filling in the questionnaires ('first questionnaires, then laboratory') as stated in the methods section (page 7, lines 25-27). We will be able to assess whether potential changes in quality of life are associated with changes in thyroid function.

- Please explain the rationale for choosing a 150 mcg dose. It seems high, given that an amount of > 100 mcg is unlikely to be used for SCH. Also, a dose of 150 mcg for a patient's weight < 50 mg is different than that of 150 mcg for a patient weighing > 50 mcg.

This study does not focus solely on participants with SCH. Also participants with overt hypothyroidism in their medical history can be included. Therefore, a daily amount of levothyroxine exceeding 100 ug is also accepted. However, for safety reasons, participants using > 150 ug levothyroxine per day are not eligible to participate as was advised by endocrinologists of the LUMC.

Adaptation Method- study population, page 7, line 3.

o '2. The dose of treatment is > 150 mcg levothyroxine per day (for safety reasons);'

- Consider excluding patients with recent hospitalizations and patients with severe dyslipidemia and severe depression.

We agree that these exclusion criteria are valid, however, we are not able to change the criteria as the recruitment has already commenced. GP's were able to exclude patients when they felt study participation could harm their patient. These reasons will be collected as well. Depression and recent hospitalization could be one of these reasons.

Adaptation Methods - study setting, page 6, lines 23-24.

o 'GPs may exclude patients when they feel study participation could harm their patient. Reasons for exclusion are collected.'

- How will adherence to levothyroxine assess at baseline and during the study?

GPs select patients when using levothyroxine at a stable dosage ≥ 1 year. With corresponding prescriptions of levothyroxine in the year prior to inclusion, these patients are considered to be adherent. Adherence will not be assessed with questionnaires during the study.

Adaptation Methods- study phases and interventions, page 8, line 23-25 (extra paragraph).

o 'Adherence Participants are contacted by the research nurse by telephone and/or by mail or email prior to each measurement and in case of missing data, to ensure data quality and to promote participant retention and complete follow-up.'

- Please explain the rationale for using the chosen measures. Why was ThyPRO39 chosen? How will symptoms be assessed during the discontinuation phase? And what are the changes in symptoms that will triggered stopping the trial for an individual patient?

We have chosen ThyPRO39 because it is a questionnaire covering a comprehensive range of thyroid-related quality of life issues and considerations spanning somatic, psychological and social domains. The ThyPRO has been nominated as the recommended method for assessing health-related quality of life in benign thyroid disease [Wong, C.K. et al., A systematic review of quality of thyroid-specific health-related quality-of-life instruments recommends ThyPRO for patients with benign thyroid diseases. J Clin Epidemiol, 2016. 78: p. 63-72].

Measuring ThyPRO39 at baseline, 6 weeks after the start of the discontinuation phase and at the end of the discontinuation phase, will give insight to symptoms that may have triggered stopping the discontinuation phase. Participants stopping the discontinuation phase do not necessarily also stop the trial. After stopping the discontinuation phase, they will be asked to attend the final visit (1 year after starting the discontinuation phase). All adverse events will be monitored. When participants or their GP indicate that there are complaints during discontinuation, these complaints will be assessed by the research nurse.

Adaptation Methods – descriptive study data and outcome measures, page 8, line 28 and page 9, line 11-12

o Addition of Appendix 2: explanation of clinical relevance of chosen efficacy outcomes.

Adaptation Methods -safety and monitoring, page 12, line 19-20.

o 'If complaints are reported during the discontinuation phase, these will be assessed by the research nurse.'

- Will cardiovascular events be assessed as a safety outcome during the trial?

Participants and GPs are asked at every step during the study to report any adverse events (AEs) or SAEs. During the study AEs and serious adverse events (SAEs) will be monitored, including

cardiovascular events. All SAEs will be reported to the accredited Medical Ethics Committee and to the Data Safety Monitoring Board.

Adaptation Methods -safety and monitoring, page 12, line 19-20.

o 'All adverse events (including cardiovascular events) reported spontaneously by the participant or their GP are recorded and followed up.'

- In the analysis, please expand on the subgroup of interest and how the heterogeneity of effect will be assessed.

Heterogeneity of treatment effect can indeed be assessed with subgroup analysis. However, this study is not powered to perform such an analysis. Also, although common, traditional subgroup analysis typically provides very limited information regarding variability in treatment effects (Angus & Chang, JAMA 2021). Therefore, we have chosen to perform univariable and multivariable logistic regression to assess factors that are associated with successful discontinuation of levothyroxine treatment.

- Is the calculated sample size large enough to estimate differences for other outcomes?

No formal sample size calculation was performed on secondary endpoints. The sample size calculation is based on the primary endpoint. For reviewer only: secondary endpoint proportion of participants that achieved substantial dose lowering can be estimated with the calculated sample size. Potentially with a larger margin of error. No effect on ThyPRO and EQ5D is expected at T52, since fT4 is within reference range at T52. The reflection of participants on their decision could be calculated with a proportion as well, since current sample size is based on a proportion of 50%, with is the largest sample size, the sample size is large enough to estimate other proportions (larger or smaller than 50%).

- Several observational discontinuation studies found that thyroid parenchyma ultrasound appearance strongly predicted discontinuation. Is this variable considered in this study? If not, why?

Thank you for your suggestion. Since our study is conducted in general practices, it is not feasible to include thyroid parenchyma ultrasound appearance as an outcome parameter. We were not aware of studies where ultrasonic features could significantly predict failure of discontinuation of levothyroxine and were included in the decision tree model.

VERSION 2 – REVIEW

REVIEWER	Brito, Juan Mayo Clinic
REVIEW RETURNED	20-Mar-2023
GENERAL COMMENTS	My concerns have been addressed thanks.

VERSION 2 – AUTHOR RESPONSE